# Untargeted high-resolution paired mass distance data mining for retrieving general chemical relationships

Miao Yu [1] & Lauren Petrick [1,2 ✉]

Untargeted metabolomics analysis captures chemical reactions among small molecules. Common mass spectrometry-based metabolomics workflows first identify the small molecules significantly associated with the outcome of interest, then begin exploring their biochemical relationships to understand biological fate or impact. We suggest an alternative by which general chemical relationships including abiotic reactions can be directly retrieved through untargeted high-resolution paired mass distance (PMD) analysis without a priori knowledge of the identities of participating compounds. PMDs calculated from the mass spectrometry data are linked to chemical reactions obtained via data mining of small molecule and reaction databases, i.e. 'PMD-based reactomics'. We demonstrate applications of PMD-based reactomics including PMD network analysis, source appointment of unknown compounds, and biomarker reaction discovery as complements to compound discovery analyses used in traditional untargeted workflows. An R implementation of reactomics analysis and the reaction/PMD databases is available as the pmd package.

[1] Department of Environmental Medicine and Public Health, Icahn School of Medicine at Mount Sinai, New York, NY 10029, USA. [2] Institute for Exposomic Research, Icahn School of Medicine at Mount Sinai, New York, NY 10029, USA. ✉email: lauren.petrick@mssm.edu

Untargeted metabolomics or nontargeted analysis using high resolution mass spectrometry (HRMS) is one of the most popular analysis methods for unbiased measurement of organic compounds[1,2]. A typical metabolomics sample analysis workflow will follow a detection, annotation, MS/MS validation, and/or standards validation process, from which interpretation of the relationships between these annotated or identified compounds can then be linked to biological pathways or disease development, for example. However, difficulty annotating or identifying unknown compounds always limits the interpretation of findings[3]. One practical solution to this is matching experimentally obtained fragment ions to a mass spectral database[4], but many compounds remain unreported/absent, thereby preventing annotation. Rules or data mining-based prediction of in silico fragment ions is successful in many applications[2,5], but these approaches are prone to overfitting the known compounds, leading to false positives. Ultimately such workflows require final validation with commercially available or synthetically generated analytical standards, which may not be available, for unequivocal identification.

Potential molecular structures could be discerned using biochemical knowledge, through the integration of known relationships between biochemical reactions (e.g., pathway analysis)[3]. Such methods are readily used to annotate compounds by chemical class. For example, the referenced Kendrick mass defect (RKMD) was able to predict lipid class using specific mass distances for lipids and heteroatoms[6], and isotope patterns in combination with specific mass distances characteristic of halogenated compounds such as +Cl/−H, +Br/−H were used to screen halogenated chemical compounds in environmental samples[7]. For these examples, known relationships among compounds were used to annotate unknown compounds, as a complementary approach to obtaining compound identifications.

The most common relationships among compounds are chemical reactions. Substrate-product pairs in a reaction form by exchanging functional groups or atoms. Almost all organic compounds originate from biochemical processes, such as carbon fixation[8,9]. Like base pairing in DNA[10], organic compounds follow biochemical reaction rules, resulting in characteristic mass differences between the paired substrates and their products. Here, we build on our paired mass distance (PMD) concept[11], that reflects such reaction rules by calculating the mass differences between two compounds or charged ions. By expanding the PMD framework, it can be used to extract biological inference without identifying unknown compounds.

Exploiting mass differences for compound identification is not new. Mass distances have been used to reveal isotopologue information when peaks show a PMD of 1 Da[12], identifying adducts from the same compound[11] such as PMD 22.98 Da between adducts [M+Na]$^+$ and [M+H]$^+$, or adducts formed via complex in-source reactions[13] from mass spectrometry data. Such between-compound information has also been used to make annotations of unknown compounds[4,14], to classify compounds[15], or to perform pathway-independent metabolomic network analysis[16]. However, these calculations of PMD were used to identify the compounds or pathways and ultimately facilitate interpretations of the relationships between these predefined important compounds. Here, we propose that PMD can be used directly, skipping the step for annotation or identification of individual compounds, to aggregate information at the reaction level, called "PMD-based Reactomics". Noticing "reactomics" has been used in previous studies based on chromatic patterns[17] or NMR spectroscopy[18], reactomics in this work will actually be PMD-based reactomics.

HRMS can directly measure PMDs with the mass accuracy needed to provide reaction level specificity. Therefore, HRMS has

the potential to be used as a reaction detector to enable reaction level study investigations. Here, we use multiple databases and experimental data to provide a proof-of-concept for using mass spectrometry in PMD-based reactomics. We also discuss potential applications such as PMD network analysis, biomarker reaction discovery, and source appointment of unknown compounds. We envision that these applications will reveal the measurable reaction level changes without the need to assign molecular structure to unknown compounds. Though the applications demonstrated here focused on biological processes, databases or reactions, abiotic reactions such as photochemical or pyrolysis reactions could also be studied using PMD-based reactomics as long as the compounds can be measured by HRMS.

## Results and discussion
Definition of concepts in PMD-based reactomics, qualitative, and relative quantitative PMD analysis are provided in the "Methods" section.

**PMD network analysis.** Using the proposed PMD network analysis (see "Methods" section and Supplementary Methods for details), we can identify metabolites associated with a known biomarker of interest. In fact, PMD network analysis can also be used in combination with classic identification techniques to enhance associated networks with targeted biomarkers. As a proof of concept, we re-analyzed data from a published study to detect the biological metabolites of exposure to Tetrabromobisphenol A (TBBPA) in pumpkin[19,20] using a local, recursive search strategy (see Fig. 1). Using TBBPA as a target of interest, we searched for PMDs linked with the debromination process, glycosylation, malonylation, methylation, and hydroxylation, which are phase II reactions (e.g., primary metabolites) found in the original paper. Using this PMD network analysis, we identified 22 unique $m/z$ ions of potential TBBPA metabolites, confirmed by the presence of brominated isotopologue mass spectral

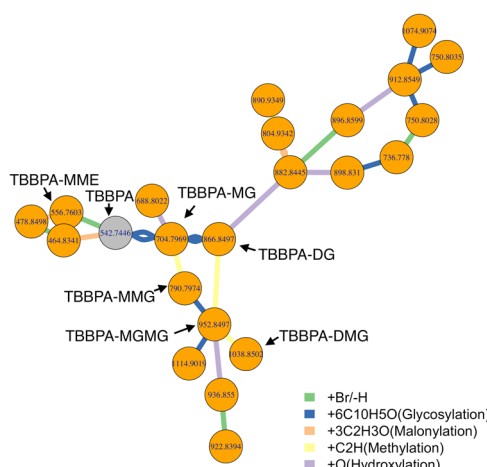

**Fig. 1 Metabolites in pumpkin seedlings' root samples following exposure to TBBPA.** Untargeted metabolomics data generated by Hou et al.[19] was re-analyzed using recursive PMD network analysis. First, the original peak list was screened to retain all of the potential brominated peaks based on mass defect analysis. Then, metabolite nodes were connected based on PMD reactions identified in the original study including 77.91 Da for debromination, 162.05 Da for glycosylation, 86.00 Da for malonylation, 14.02 Da for methylation, and 15.99 Da for hydroxylation. Finally, edges between two nodes were defined by Pearson's correlation coefficients > 0.6. Recursive PMD network analysis identified 23 metabolites, including the seven labeled metabolites of TBBPA previously reported[19].

patterns (Supplementary Fig. 1). This total was 15 more than the seven unique ions that were described in the original publication. Such a network was built based on the experimental data and our local fast recursive search algorithms as shown in Supplementary Methods: PMD network analysis. As shown in Fig. 1, most of the potential metabolites of TBBPA were found as higher-generation TBBPA metabolites, which are too computationally intensive to be identified using in silico prediction and matching protocols[21].

Similar applications and methodology have been reported for fourier transform mass spectrometry data to build a metabolic network in biological samples[22,23]. However, based on accuracy analysis (see Supplementary Results and Discussion: PMD requires HRMS), we show that quadrupole time-of-flight mass spectrometry also has the capability to perform such analysis for small molecules. In addition, our analysis considers the relationship among all paired ions to screen all of the possible metabolites of metabolites, while the previous study only considers the peaks correlated with the parent compounds[22]. Furthermore, PMD-based reactomics as described here, can be implemented beyond biochemical analysis to explore abiotic reactions such as photochemical or pyrolysis reactions. Using the same workflow, network analysis can be used to track the environmental/abiotic fate of chemical compounds as long as their corresponding PMDs show high frequency in the data (such a feature is also available in the pmd package).

**Source appointment of unknown compounds**. When an unknown compound is identified as a potential biomarker, determining whether it is associated with endogenous biochemical pathways or exogenous exposures can provide important information toward identification. High frequency PMDs from Human Metabolome Database (HMDB) and Kyoto Encyclopedia of Genes and Genomes (KEGG) are dominated by reactions with carbon, hydrogen, and oxygen suggesting links to metabolism pathways (See Supplementary Tables 1, 2, and 3). Therefore, if an unknown biomarker is mapped using a PMD network, connection to these high frequency PMDs would suggest an endogenous link. However, separation from this network is expected for an exogenous biomarker in which the reactive enzyme is not in the database. The exogenous compound is secreted in the parent form, or can undergo changes in functional groups such as during phase I and phase II xenobiotic metabolism processes. In this case, endogenous and exogenous compounds should be separated by their PMD network in samples.

Topological differences in PMD networks for endogenous and exogenous compounds were explored using compounds from The Toxin and Toxin Target Database (T3DB)[24]. As shown in Fig. 2, the PMD network of compounds was generated based on the top ten high frequency PMDs of 255 endogenous compounds with 223 unique masses, and 705 exogenous compounds with 394 unique masses and carcinogenic 1, 2A, or 2B classifications. Most endogenous compounds (Fig. 2, orange) were connected into a large network, while the exogenous compounds' networks were much smaller (Fig. 2, blue). Interestingly, most carcinogenic compounds were not connected by high frequency PMDs. Expanding this beyond just carcinogenic compounds, we randomly sampled 255 exogenous compounds from a total of 2491 exogenous compounds available in T3DB, and built a PMD network with the top ten high frequency PMDs of those 510 compounds (255 exogenous compounds and 255 endogenous compounds). This step was repeated 1000 times, and the average degree of connection with other nodes was calculated as 4.5 (95% Confidence Interval, CI [4.3, 4.8]) for endogenous compounds and 1.7 (95% CI [1.2, 2.2]) for exogenous compounds.

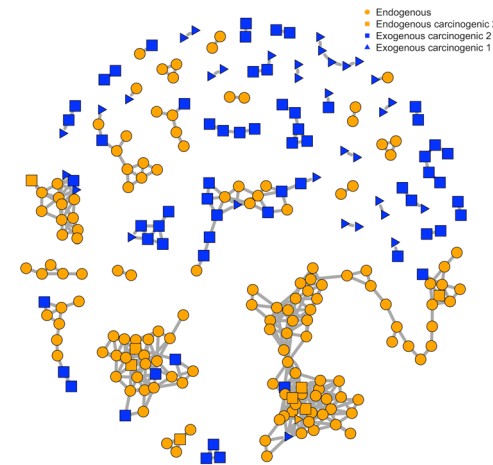

**Fig. 2 PMD network of endogenous and exogenous compounds from T3DB.** PMD analysis was performed on all 255 endogenous compounds (with 223 unique chemical formula) and 705 exogenous carcinogens (with 394 unique chemical formula) from the T3DB database. PMD network analysis was then performed using the top ten frequency PMDs from the 617 unique formulas. 235 compounds with unique chemical formulas did not have linkages, and were removed. The remaining 157 endogenous (orange squares and circles) and 95 exogenous compounds (blue squares and triangles) were used to generate the network.

Similar findings were observed for known compounds. For demonstration, we selected caffeine, glucose, bromophenol, and 5-cholestene as well characterized chemicals that are commonly observed with mass spectrometry, and paired them with other metabolites in the KEGG reaction database using the top ten high frequency PMDs from Supplementary Table 1. As shown in Fig. 3, different topological properties (e.g., number of nodes, average distances, degree, communities, etc.) of compounds' PMD network were observed for each selected target metabolite. Endogenous compounds such as glucose or 5-cholestene were highly connected (average degree of node is 3.4 and 3.2, respectively) while exogenous compounds such as caffeine and bromophenol have more simple networks (average degree of node is 2.2 and 2.4, respectively). Further, the average PMD edge numbers between all nodes (edges end-to-end) in glucose and 5-cholestene networks are 9.7 and 6.6, respectively, while the average PMD edge numbers for caffeine and bromophenol are 3.3 and 1.8, respectively. Larger average PMD edge numbers mean a complex network structure with lots of nodes, while smaller average PMD edge numbers mean a simple network structure with a few nodes. Based on these estimates, we proposed that unknown metabolites with average network node degree more than three would be likely endogenous compounds. Similarly, if the unknown compound belongs to a network with longer average PMD edge numbers, such compounds might also be of endogenous origin. The R code to generate compound networks for any compound in the KEGG database is available in the Supplementary Methods.

**Biomarker reactions**. PMD-based reactomics can be used to discover biomarker "reactions" instead of biomarker "compounds". Unlike typical biomarkers that are a specific chemical compound, biomarker reactions contain all peaks within a fixed PMD relationship and correlation cutoff. Thus, relative quantitative PMD analysis (see "Methods" section for details) can be used to determine if there are differences between groups (e.g., control or treatment, exposed or not-exposed) on a reaction level. Such differences are described as a biomarker "reaction".

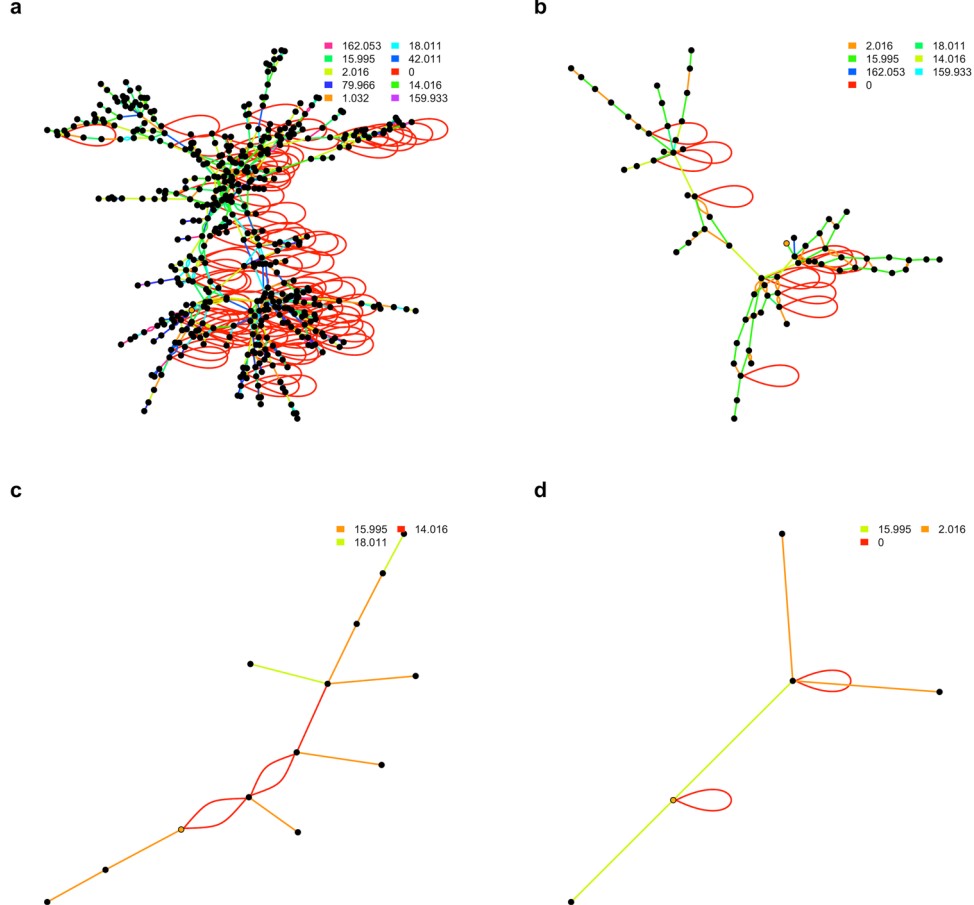

**Fig. 3 PMD networks for selected compounds from the KEGG reaction database.** For each compound: glucose (**a**), 5-cholestene (**b**), caffeine (**c**), and bromophenol (**d**), networks were generated using the ten top frequency PMDs identified in KEGG. For each network, the selected compound is indicated with an orange node. The associated metabolites are indicated by a black node, with edges colored based on the PMD of the relationship between nodes. Double edges from the same pairs or self-looped edges indicate isomers-related relationships.

We used publicly available metabolomics data (MetaboLight ID: MTBLS28) collected on urine from a study on lung cancer in adults[25]. Four peaks out of 1807 features from 1005 blood samples (469 cases and 536 controls) generated the quantitative responses of PMD 2.02 Da. This biomarker reaction (e.g. +2H from our annotated database) was significantly decreased in case samples compared with the control group (*t*-test, $p < 0.05$, see Fig. 4). The original publication associated with this dataset did not report any molecular biomarker associated with this reaction[25], or the metabolites linked with this reaction, suggesting that relative quantitative PMD analysis offers additional information on biological differences between the groups on the reaction level that may be lost when focused on analysis at the chemical level. PMD-level investigations directly reduce the high dimensional analysis typically performed on a peaks or features level into low dimensional analysis on the chemical reaction level with explainable elemental compositions. Furthermore, these results suggest that follow-up analysis in this population should include targeted analysis of proteins or enzymes linked with +2H changes.

In summary, we provide the theoretical basis and empirical evidence that high resolution mass spectrometry can be used as a reaction detector through calculation of high resolution paired mass distances and linkage to reaction databases such as KEGG. PMD-based reactomics, as a new concept in bioinformatics, can be used to find biomarker reactions or develop PMD networks. The major limitation of PMD-based reactomics analysis is that

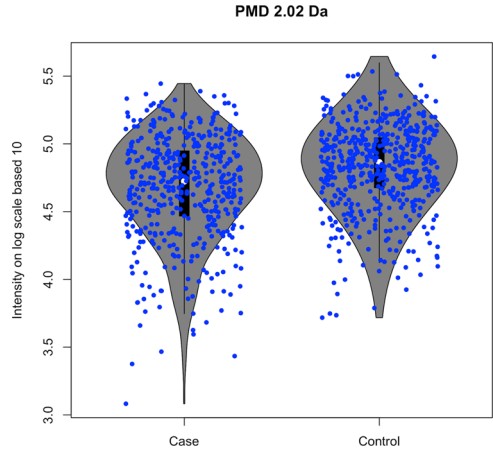

**Fig. 4 PMD differential analysis (*t*-test, *p*-value < 0.05) identifies PMD 2.02 Da as a potential biomarker reaction for lung cancer.** Data from the public MetaboLights repository (MTBLS28 dataset) were analyzed with relative quantitative PMD analysis. PMD pairs of 2.02 Da with consistent intensity ratios (RSD% < 30%) across all samples were selected. Then, the total intensity of ions with a PMD 2.02 Da was calculated in each sample and compared between cases and controls.

mass spectrometry software is designed for analysis of compounds instead of reactions. In this case, the uncertainty in PMD measurements can not be captured directly from the instrument, and instead are calculated after data acquisition. Furthermore, while PMD-based reactomics can be applied to analyze environmental samples, the absence of publicly available reaction databases for environmental processes currently limits PMD-based applications to the analysis of biological samples. Nevertheless, PMD-based reactomics techniques provide information on biological changes for new biological inferences, that may not be observed through classic chemical biomarker discovery strategies.

## Methods

**Definitions**. We first define a reaction PMD ($PMD_R$) using a theoretic framework. Then we demonstrate how a $PMD_R$ can be calculated using KEGG reaction R00025 as an example (see Eq. (1)). There are three KEGG reaction classes (RC00126, RC02541, and RC02759) associated with this reaction, which is catalyzed by enzyme 1.13.12.16.

$$\text{Ethylnitronate} + \text{Oxygen} + \text{Reduced FMN} <=> \text{Acetaldehyde} + \text{Nitrite} + \text{FMN} + \text{Water} \quad (1)$$

In general, we define a chemical reaction ($PMD_R$) as follows Eq. (2):

$$S_1 + S_2 + \ldots + S_n <=> P_1 + P_2 + \ldots + P_m \, (n \geq 1, m \geq 1), \quad (2)$$

where $S$ means substrates and $P$ mean products, and $n$ and $m$ the number of substrates and products, respectively. A PMD matrix [3] for this reaction is generated:

$$
\begin{array}{ccccc}
 & S_1 & S_2 & \ldots & S_n \\
P_1 & |S_1 - P_1| & |S_2 - P_2| & \ldots & |S_n - P_1| \\
P_2 & |S_1 - P_1| & |S_2 - P_2| & \ldots & |S_n - P_2| \\
\ldots & \ldots & \ldots & \ldots & \ldots \\
P_m & |S_1 - P_m| & |S_2 - P_m| & \ldots & |S_n - P_m|
\end{array}
\quad (3)
$$

For each substrate, $S_k$, and each product, $P_i$, we calculate a PMD ($|S_n - P_m|$).

Assuming that the minimum PMD would have a similar structure or molecular framework between substrate and products, we select the minimum numeric PMD for each substrate as the substrate PMD ($PMD_{Sk}$) of the reaction (Eq. (4)).

$$PMD_{Sk} = \min(|S_k - P_1|, |S_k - P_2|, \ldots, |S_k - P_m|)(1 <= k <= n) \quad (4)$$

Then, the $PMD_R$, or overall reaction PMD, is defined as the set of substrates' PMD(s) (Eq. (5)):

$$PMD_R = \{PMD_{S1}, PMD_{S2}, \ldots, PMD_{Sn}\} \quad (5)$$

For KEGG reaction R00025, $S_1$ is ethylnitronate, $S_2$ is oxygen, $S_3$ is reduced FMN, $P_1$ is acetylaldehyde, $P_2$ is nitrite, $P_3$ is FMN, $P_4$ is water, $n = 4$, and $m = 3$. A PMD matrix (6) for this reaction can be seen below (absolute value calculations indicated in italics, corresponding formula matrix can be found in Supplementary Note 1), where we define $PMD_{Ethylnitronate} = 27.023$ Da, $PMD_{Oxygen} = 12.036$ Da, and $PMD_{Reduced\ FMN} = 2.016$ Da.

|  | Ethylnitronate | Oxygen | Reduced FMN |
|---|---|---|---|
| Acetaldehyde | 29.998 Da $|74.0242 - 44.0262|Da$ | 12.036 Da $|31.9898 - 44.0262|Da$ | 414.094 Da $|458.1202 - 44.0262|Da$ |
| Nitrite | 27.024 Da $|74.0242 - 47.0007|Da$ | 15.011 Da $|31.9898 - 47.0007|Da$ | 411.120 Da $|458.1202 - 47.0007|Da$ |
| FMN | 382.080 Da $|74.0242 - 456.1046|Da$ | 424.115 Da $|31.9898 - 456.1046|Da$ | 2.016 Da $|458.1202 - 456.1046|Da$ |
| $H_2O$ | 56.014 Da $|74.0242 - 18.0105|Da$ | 13.979 Da $|31.9898 - 18.0105|Da$ | 440.110 Da $|458.1202 - 18.0105|Da$ |

$$(6)$$

In our example, there are three $PMD_R$ calculated from three $PMD_S$: $PMD_R$ is 27.023 Da, which is equivalent to the mass difference between two carbon atoms and three hydrogen atoms: $PMD_R$ is 12.036 Da for the additions of two carbon atoms and four hydrogen atoms and loss of one oxygen atom: and $PMD_R$ is 2.016 Da for the addition of two hydrogen atoms. However, other reactions may have multiple $PMD_S$ that generate the same $PMD_R$ value, such as certain combination reactions or replacement reactions. In this case, only one value will be kept as reaction PMD as long as it is the minimum PMD for all of the involved substrates. In addition, each $PMD_R$ has two notations. One is shown as an absolute mass difference of the substrate-product pairs' exact masses or monoisotopic masses with unit Da. Another notation is using elemental compositions as the differences between two chemical formulas. Here, we describe it as an elemental composition instead of chemical formula, because it also describes the gain and loss of elements, and therefore the neat mass change. In our example reaction, the $PMD_R$ can also be written as +2C3H, +2C4H/−O, and +2H, respectively. This elemental composition can be linked to known chemical processes retrieved from a reaction database, i.e., KEGG. For example, +2H represents the elemental composition change of a reaction involving a double bond breaking such as KEGG example RC00126, and +2C3H indicates reaction with nitronate monooxygenase (EC:1.13.12.16) or reaction class RC02541. However, some elemental compositions, such as +2C4H/−H in our example, might not have a clear mechanism (e.g., no

suggested KEGG reaction selection). By this definition, $PMD_R$ can be generated automatically in terms of elemental compositions or mass units in Da.

We used these definitions to establish reference databases of PMDs. We used KEGG as a "reaction database" representing common reactions in human endogenous pathways, and we used HMDB[19] as the "compound database" representing common reactions between chemicals measured in human biofluids (see Supplementary Methods for data mining details).

**Qualitative and relative quantitative PMD analysis**. PMD can be determined in biological or environmental samples from peaks observed in mass spectrometry. Mathematically, a PMD of uncharged compounds is equivalent to the PMD of their charged species observed with a mass spectrometer, as long as both compounds share the same adducts, neutral losses, and charges. In example reaction [1], reduced FMN has a monoisotopic mass of 458.1203 Da, while FMN has a monoisotopic mass of 456.1046 Da. Spectra from HMDB[19] showed that common ions for reduced FMN and FMN using liquid chromatography (LC)-HRMS in negative mode are typically $[M-H]^-$ with $m/z$ 457.1124 and 455.0968, respectively. The mass distance of the monoisotopic masses is 2.016 Da and the mass distance of the observed adducts is also 2.016 Da. In cases such as this, mass spectrometry can be used to detect the PMD of paired compounds, but only for HRMS (see Supplementary Results and discussion: redundant peaks and fragments in PMD-based reactomics).

In addition to qualitative analysis, peaks that share the same PMD can be summed and used as a relative quantitative group measure of that specific "reaction" in the sample, thereby providing a description of chemical reaction level changes across samples without annotating individual compounds. We define two types of PMD across samples: static PMD in which intensity ratios between the pairs are stable across samples, and dynamic PMD in which the intensity ratios between pairs change across samples. Only static PMDs, those with similar instrument response, can be used for relative quantitative analysis (see Supplementary Table 4 for theoretical example). Similar to other nontargeted analysis[20], a relative standard deviation (RSD) between quantitative pair ratios <30% and a high correlation between the paired peaks' intensity (>0.6) are suggested to be considered a static PMD.

## Data availability

All of the dataset (Supplementary Data 1 for HMDB, Supplementary Data 2 for KEGG, Supplementary Data 3 for T3DB and Supplementary Data 4 for MTBLS28) and reproducible R script (Supplementary Data 5) for all of the figures, tables and calculations are supplied in Supplementary Information.

## Code availability

An R implementation of PMD-based reactomics analysis and the reaction/PMD databases is available as the pmd package (https://yufree.github.io/pmd/). The stable version of the pmd package can also be accessed from CRAN (https://cran.r-project.org/web/packages/pmd/index.html).

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

## Acknowledgements
This research was financially supported by NIEHS grants P30ES23515, 1U2CES030859, R21ES030882, and R01ES031117. We thank the Sanchez group (Gordon Luu, Alanna Condren, Jessica Cleary, Katherine Zink, Cynthia Grim, and Laura Sanchez) for their comments in open review for the preprint of this manuscript.

## Author contributions
Miao Yu: Conceptualization, software development, data curation, visualization, writing-original draft, writing-review and editing; L.P. writing review and editing, supervision, project administration, funding acquisition. All authors read, reviewed, and accepted the final manuscript.

## Competing interests
The authors declare no competing interests.
