## [Peer Review File · Communications Chemistry]

Reviewers' comments:

Reviewer #1 (Remarks to the Author):

The manuscript "Reactomics: using mass spectrometry as a reaction detector" by Miao Yu and Lauren Petrick describes the concept of mining specific m/z related to biotransformations of specific molecules in high resolution full mass scan data. As the authors point out, the knowledge of these existing relationships is not novel, but they were able to conceptualize it well and apply it by compiling bioreactions in KEGG, by re-analyzing data and by constructing networks using their proposed workflow. This research is a valuable and important approach considering that the current methods of matching ions at databases are deprived of any chemical intelligence. Even though the authors offer a R package which is accessible for researchers with bioinformatics skills, overall the manuscript is missing details that would allow colleagues to use it as a reference and use this approach to analyze their data. To improve the communication and understanding of this manuscript here are comments for the authors to consider.

1. The supplementary files should include the data mentioned in the manuscript and used for the figures and calculations.
2. There is a whole field that uses mass spectrometry to monitor medicinal chemistry reactions for synthetic purposes. It seems to me that it would be more appropriate to use terms such as bioareactions and bioreactome. Even though reactions in bulk or inside organisms share mechanisms, the proposed approach and applications is focused on living organisms.
3. Line 72 "However, these calculations of PMD were used to identify compounds or pathways in ultimately facilitate interpretations..." Did the authors mean "...and ultimately facilitate interpretations ..."?
4. Matrix between lines 118 and 119: please add the monoisotopic masses of the compounds mentioned so that the reader can follow the matrix calculations.
5. Lines 126-127: "In this case, only one value will be kept as reaction PMD.". Isn't it "reaction minimum PMD"?
6. Lines 130-131: "Here, we describe it as an elemental composition instead of chemical formula, because it also describes the gain and loss of elements, ..." Adding this information to the matrix (lines 118 and 119) as a supplementary materials would be useful for understanding.
7. Overall, figure captions are too short and uncomplete. They should be self-explaining. Figure 2: How many compounds without linkage have been eliminated? Why just the top 10 high frequency PMDs network?
8. Figure 3: How the compounds have been selected? By pathway? Why limited to 10 top frequent PMDs? Shouldn't it be better to consider most meaningful PMDs related to such pathways?
9. In line 267 the authors refer to "quantitative PMD analysis" and refer to Figure 4. In analytical chemistry quantitative means absolute amounts, but it seems the meaning here is the quantities related to the distance among m/z. Would the authors please clarify that.
10. Figure 4: The authors state that there is a significant difference between case and control, but this is not clear when observing the amount of variation and distribution of the data points. So more explanation on how the data was analyzed and the data itself (as supplementary file) is needed. The caption is not informative at all. What about the other reactions in this dataset, no changes were observed?
11. The list of 10 high frequency KEGG reaction PMDR is too short. If this manuscript is to be used as a useful reference for the approach, it should be more comprehensive (excel table for example).

Reviewer #2 (Remarks to the Author):

Yu and Petrick present a generic approach to extracting reaction information from non-annotated, untargeted high-resolution mass spectrometry metabolome data. This is an important and useful approach, based on a straightforward concept, and the authors nicely illustrate how to make use of the information, for instance to define biomarkers based on differential detection of reaction

markers (diagnostic paired mass differences). Their principled approach, which can comprehensively identify the relevant reaction-specific mass differences from large reaction databases, generalises earlier work in a powerful way and is technically sound and convincing.

The authors correctly state that the underlying concept is not new. However, among the cited precedents, I am missing the most similar early articles (Breitling et al. 2006 PMID: 24489532, 17064801), which already presented some closely related ideas, in particular the use of paired mass differences to organise of metabolome data without prior knowledge of metabolite identities; e.g., compare Figure 1 of the manuscript to Figure 2 in Breitling et al. A brief discussion of the relevant advances compared to this earlier work would be helpful.

I am also not entirely convinced by the choice of name for the proposed methodology: "reactomics" has already been used before (e.g., Kolusheva et al. 2012 PMID: 22778601 "A novel "reactomics" approach for cancer diagnostics"; Sundekilde et al. 2018 "Real-time monitoring of enzyme-assisted animal protein hydrolysis by NMR spectroscopy – An NMR reactomics concept"), for related, but not identical concepts. This could lead to avoidable confusion.

Response to Reviewers' Comments

We thank the reviewers for their insightful comments. We have substantially revised the manuscript on their basis, and have detailed the changes made in response to each comment below. We have also adjusted the manuscript and supplementary information structure in line with the requirement for Communications Chemistry.

Reviewer #1:

The manuscript "Reactomics: using mass spectrometry as a reaction detector" by Miao Yu and Lauren Petrick describes the concept of mining specific m/z related to biotransformations of specific molecules in high resolution full mass scan data. As the authors point out, the knowledge of these existing relationships is not novel, but they were able to conceptualize it well and apply it by compiling bioreactions in KEGG, by re-analyzing data and by constructing networks using their proposed workflow. This research is a valuable and important approach considering that the current methods of matching ions at databases are deprived of any chemical intelligence.

Even though the authors offer a R package which is accessible for researchers with bioinformatics skills, overall the manuscript is missing details that would allow colleagues to use it as a reference and use this approach to analyze their data. To improve the communication and understanding of this manuscript here are comments for the authors to consider.

1. The supplementary files should include the data mentioned in the manuscript and used for the figures and calculations.

Reply: *We agree and have attached all of the data and reproducible R script for all of the figures and calculation as supplementary information. Since the databases (HMDB and T3DB) might be updated in the future, we have also provided the databases that we used for HMDB and T3DB. As KEGG is routinely updated, we directly release the code to access the database online in real time. As a consequence, the final results might be slightly different if the reviewers re-run the code to generate the figures.*

Here is the related files list found in the supplementary information:

HMDB.csv: All of the compounds in HMDB database, accessed on 2019-10-02.

Keggrall.csv: KEGG PMDs annotation database, accessed on 2020-05-04.

T3db.csv: All of the compounds in T3DB database, accessed on 2018-10-10.

MTBLS28posmzrt.csv: Peaks list from MTBLS28 project.

RSI.r: R code to reproduce all of the figures, tables and calculation in this study.

2. There is a whole field that uses mass spectrometry to monitor medicinal chemistry reactions for synthetic purposes. It seems to me that it would be more appropriate to use terms such as bioactions and bioactome. Even though reactions in bulk or inside organisms share mechanisms, the proposed approach and applications is focused on living organisms.

Reply: *We agree that reactomics can be used in many fields. As such, we have clarified the name to 'PMD-based reactomics' to reflect that its use is not limited to biological applications. For example, an important application for PMD-based reactomics is in the field of environmental sciences. Environmental chemical reactions can be either biologically related (e.g. methylation or hydroxylation) or driven by environmental factors such as photochemical or pyrolysis reactions. Unlike KEGG, databases for those reactions are not readily available, which limits our ability to demonstrate the potential of reactomics in environmental processes. However, the concept can be readily applied to all possible chemical reactions as long as they can be detected using mass spectrometry. We have added or modified related sentences to clarify this in the manuscript including at the end of introduction (lines 86-88), in the discussion section of TBBPA metabolites (lines 118-120), as well as in the abstract.*

3. Line 72 “However, these calculations of PMD were used to identify compounds or pathways in ultimately facilitate interpretations...” Did the authors mean “...and ultimately facilitate interpretations ...”?

Reply: *Thanks, this has been corrected (line 72).*

4. Matrix between lines 118 and 119: please add the monoisotopic masses of the compounds mentioned so that the reader can follow the matrix calculations.

Reply: *We have added the monoisotopic masses to matrix 2 [M2]*

5. Lines 126-127: “In this case, only one value will be kept as reaction PMD.”. Isn't it “reaction minimum PMD”?

Reply: *Yes, this was clarified as suggested (line 251).*

6. Lines 130-131: “Here, we describe it as an elemental composition instead of chemical formula, because it also describes the gain and loss of elements, ...” Adding this information to the matrix (lines 118 and 119) as a supplementary materials would be useful for understanding.

Reply: *This has been added to the supplementary information (Supplementary Notes, MS1)*

7. Overall, figure captions are too short and incomplete. They should be self-explaining.

Reply: We agree, and have gone through and added additional detail to each figure caption.

Figure 2: How many compounds without linkage have been eliminated? Why just the top 10 high frequency PMDs network?

Reply: 235 unique chemical formulas have been removed due to no linkage, and this has now been added to the Figure 2. caption. As we mentioned in the supplementary information (Section data mining lines 117-120), the 10 highest frequency values covered 5448 of the 9200 KEGG reactions. Therefore, it can be assumed that a similar enrichment of PMD would also occur in other biochemical databases and real samples. Further, a demonstration of a cut-off determination of the top 10 PMDs for the T3DB analysis in Fig 2 is shown below and now provided in the reproducible code of supplemental information.

We plotted the number of unique chemical formulas as a function of PMD frequency cut-off for endogenous (red), exogenous (blue), and both endogenous and exogenous compounds (black) from the 617 T3DB compounds. As can be seen, the endogenous compounds show a different increasing rate of unique formula with increasing PMD cut-off (N) than the exogenous compounds. In order to select a cut-off, a balance must be met between too large of a cut-off, where all compounds will be connected to one network without separation and result in the inclusion of random PMDs among compounds, or too small of a cut-off, where the connections among compounds with chemical reaction meanings are lost. In the figure below, after the top 13 PMDs, the number of unique formulas for endogenous compounds begin to plateau. Similarly, the first plateau for the number of unique formulas for exogenous compounds occurs after the top 12 PMDs. Combining the endogenous and exogenous compounds together, the total unique formula numbers start flattening after 12 PMDs. These first plateaus suggest a cut-off estimate of ~10 PMDs is appropriate, which is also consistent with the top 10 KEGG PMDs used in the related discussion.

We have also supplied the code that allows generation of the network from 1 up to 50 top PMDs for independent validation (see line 120-141 in RSI.r).

As recommended, the new caption for figure 2 has been changed to:

Figure 2. PMD network of endogenous and exogenous compounds from T3DB. PMD analysis was performed on all 255 endogenous compounds (with 223 unique chemical formula) and 705 exogenous carcinogens (with 394 unique chemical formula) from the T3DB database. PMD network analysis was then performed using the top 10 frequency PMDs from the 617 unique formulas. 235 compounds with unique chemical formulas did not have linkages, and were removed. The remaining 157 endogenous (red squares and circles) and 95 exogenous compounds (blue squares and triangles) were used to generate the network.

8. Figure 3: How the compounds have been selected? By pathway? Why limited to 10 top frequent PMDs? Shouldn't it be better to consider most meaningful PMDs related to such pathways?

Reply: The compounds were selected based on their sources (endogenous or exogenous) as examples. However, any compounds in the KEGG database could be selected, and we have supplied the code to explore these other compounds in the supplementary materials (see line 291-319 in RSI.r). We limited this analysis to the top 10 PMDs to make it consistent with figure 2 and related discussion, but a different cut-off could be larger or smaller depending on the user preferences. While in some specific cases it may be more meaningful to consider known PMDs, the value of PMD-based reactomics is in untargeted analysis, for which pre-defining the active reactions in a certain system is very challenging. Therefore, we allowed the data (either samples or databases) to determine the enriched PMDs and used a frequency cut-off for selection.

The new caption will be:

Figure 3. **PMD networks for selected compounds from the KEGG reaction database.** For each compound: glucose (A), 5-cholestene (B), caffeine (C), and bromophenol (D), networks were generated using the 10 top frequency PMDs identified in KEGG. For each network, the selected compound is indicated with a red node. The associated metabolites are indicated by a black node, with edges colored based on the PMD of the relationship between nodes. Double edges from the same pairs or self-looped edges indicate isomers-related relationships.

9. In line 267 the authors refer to “quantitative PMD analysis” and refer to Figure 4. In analytical chemistry quantitative means absolute amounts, but it seems the meaning here is the quantities related to the distance among m/z. Would the authors please clarify that.

Reply: Thank you for this comment. Indeed, here the quantitative PMD analysis actually uses the total intensity of all the paired ions with the same PMD. However, due to the change of response factor between substitute and product compounds, we only selected the pairs with consistent intensity ratios ($RSD\% < 30\%$) across all of the samples for quantitative PMD. We agree that ‘relative quantitation’ more accurately reflects this analysis, and have changed the language accordingly throughout the main text.

Furthermore, the new caption for Figure 4 will be:

Figure 4. PMD differential analysis (t-test, p-value < 0.05) identifies PMD 2.02 Da as a potential biomarker reaction for lung cancer. Data from the public MetaboLights repository (MTBLS28 dataset) were analyzed with relative quantitative PMD analysis. PMD pairs of 2.02 Da with consistent intensity ratios ($RSD\% < 30\%$) across all samples were selected. Then, the total intensity of ions with a PMD 2.02 Da was calculated in each sample and compared between cases and controls.

10. Figure 4: The authors state that there is a significant difference between case and control, but this is not clear when observing the amount of variation and distribution of the data points. So more explanation on how the data was analyzed and the data itself (as supplementary file) is needed. The caption is not informative at all. What about the other reactions in this dataset, no changes were observed?

Reply: We identified 64 high frequency PMDs in the lung cancer data using PMD analysis, and have added these raw peaks list to the supplementary method (section: Code and data for the whole study). Further, the code to identify and screen the high frequency PMDs is also included.

In addition to PMD 2.02 Da, we saw several other PMDs with statistical differences between cases and controls including: PMD 0.04 Da, 21.98 Da, 26.02 Da, 1.00 Da, 0 Da, and 2.01 Da. A limitation of using data from a public database is that PMDs could not be validated. Since 21.98 Da could be a sodium adduct, and the other PMDs might come from low accuracy of MS (e.g.

0.04 Da, 2.01 Da), isomers (e.g. 0 Da), isotope (e.g. 1.00 Da) or other ions (e.g. 26.02 Da), we only included 2.02 Da for this paper to demonstrate the concept since it is also listed in the top 10 KEGG PMDs in Supplementary Table 1. However, since we released the code for this analysis, readers can apply this analysis to any dataset they preferred to check for biomarker reactions.

11. The list of 10 high frequency KEGG reaction PMDR is too short. If this manuscript is to be used as a useful reference for the approach, it should be more comprehensive (excel table for example).

Reply: Thank you for this comment. We have now included the full KEGG PMD table as supplementary material (*Keggrall.csv*) to facilitate the readers who are not familiar with R programming.

Reviewer #2 (Remarks to the Author):

Yu and Petrick present a generic approach to extracting reaction information from non-annotated, untargeted high-resolution mass spectrometry metabolome data. This is an important and useful approach, based on a straightforward concept, and the authors nicely illustrate how to make use of the information, for instance to define biomarkers based on differential detection of reaction markers (diagnostic paired mass differences). Their principled approach, which can comprehensively identify the relevant reaction-specific mass differences from large reaction databases, generalises earlier work in a powerful way and is technically sound and convincing.

The authors correctly state that the underlying concept is not new. However, among the cited precedents, I am missing the most similar early articles (Breitling et al. 2006 PMID: 24489532, 17064801), which already presented some closely related ideas, in particular the use of paired mass differences to organise of metabolome data without prior knowledge of metabolite identities; e.g., compare Figure 1 of the manuscript to Figure 2 in Breitling et al. A brief discussion of the relevant advances compared to this earlier work would be helpful.

Reply: Thanks for the comments. We missed those pieces and related contents, and have added the corresponding citations to the main text [ref 22 and 23]. Though those two papers looked similar in terms of figure, the generation process of the network is different. Breitling et al. only considered the peaks correlated with the parent compounds, while our methodology considers further relationships among all paired ions to screen all of the metabolites of metabolites (e.g. several degrees of separation).

We have added this to the main text (line 112-123)

‘Similar applications and methodology have been reported for fourier transform mass spectrometry data to build a metabolic network in biological samples^{22,23}. Based on accuracy analysis (see Supplementary Results and Discussion: PMD requires HRMS), we show that quadrupole time-of-flight mass spectrometry also has the capability to perform such analysis for small molecules. In addition, our analysis considers the relationship among all paired ions to screen all of the possible metabolites of metabolites, while the previous study only considers the peaks correlated with the parent compounds²². Furthermore, PMD-based reactomics as described here, can be implemented beyond biochemical analysis to explore abiotic reactions such as photochemical or pyrolysis reactions. Using the same workflow, network analysis can be used to track the environmental/abiotic fate of chemical compounds as long as their corresponding PMDs show high frequency in the data (such a feature is also available in pmd package).’

I am also not entirely convinced by the choice of name for the proposed methodology: “reactomics” has already been used before (e.g., Kulusheva et al. 2012 PMID: 22778601 “A novel "reactomics" approach for cancer diagnostics”; Sundekilde et al. 2018 “Real-time monitoring of enzyme-assisted animal protein hydrolysis by NMR spectroscopy – An NMR reactomics concept”), for related, but not identical concepts. This could lead to avoidable confusion.

Reply: We agree that this may cause confusion, and have re-named this concept as ‘PMD-based reactomics’ for clarification. This has been updated throughout the text. Furthermore, we have added these citations in the main text to emphasize that our definition is focused on a PMD-based analysis (line 75-77).

“Noticing ‘reactomics’ has been used in previous studies based on chromatographic patterns¹⁷ or NMR spectroscopy¹⁸, reactomics in this work will actually be PMD-based reactomics.”

REVIEWERS' COMMENTS:

Reviewer #1 (Remarks to the Author):

I would like to thank the authors for carefully addressing the comments and adding the requested data. I recommend this manuscript for publication.